# Barriers and facilitators to palliative care service utilization in Ethiopia: A qualitative systematic review, 2025

**Sadik Abdulwehab**[1]*, **Frezer Kedir**[2]

**1** School of Nursing, Wollega University, Oromia, Ethiopia, **2** School of Nursing, Jimma University, Southwest Oromia, Ethiopia

* sadikabdulwehab@gmail.com

## Abstract

### Introduction

Palliative care is a crucial component of end-stage disease management, but its utilization remains low in low- and middle-income countries, such as Ethiopia. This is due to various systemic, social, and policy barriers. Understanding these contextual factors is crucial for developing effective interventions and policy frameworks. This study aimed to explore and synthesize the barriers and facilitators to palliative care service utilization in Ethiopia using a qualitative systematic review approach.

### Methods

This review followed the Preferred Reporting Items for Systematic Reviews and Meta-Analyses (PRISMA) 2020 guidelines and the Preferred Reporting Items for Systematic Reviews and Meta-Analyses of Qualitative Synthesis (PRISMA-Q) guidelines, employing a qualitative systematic design. A comprehensive search was conducted across various databases using tailored keywords and MeSH terms up. The database was searched for every article published on palliative care services up to March 10, 2025, and was updated continuously until it was sent for publication. The data was extracted from March 11–20 and later analyzed from March 21–30, and the report generation till April 10, 2025. Thematic synthesis was used to analyze findings and the Grading of Recommendations Assessment, Development, and Evaluation Confidence in the Evidence from Reviews of Qualitative research approach was employed to assess the confidence of evidence.

### Results

Six studies met the inclusion criteria, encompassing diverse Ethiopian healthcare settings and stakeholders. Five major barriers were identified: policy and governance gaps, health system challenges, knowledge and training deficits, sociocultural and

**Data availability statement:** The data supporting this qualitative systematic review consist of a full data extraction table (Tables 1–6), which includes study characteristics, key themes, supporting quotations, and coding decisions derived from publicly available primary studies.

**Funding:** The author(s) received no specific funding for this work.

**Competing interests:** The authors have declared that no competing interests exist.

economic constraints, and poor collaboration. In contrast, five facilitators emerged: strong community and family support, intrinsic healthcare provider motivation, integration of palliative care into education, holistic care models, and stakeholder engagement. High confidence was assigned to four themes, underscoring their significance and applicability.

## Conclusion

The underutilization of palliative care in Ethiopia stems from intertwined structural, educational, and sociocultural challenges. However, promising facilitators exist that can guide policy reform and intervention design. Addressing these barriers through improved policies, workforce development, and community engagement is imperative for ensuring equitable access to quality palliative care services.

## Introduction

The International Association for Hospice and Palliative Care (IAHPC) defines palliative care as holistic, active care for individuals suffering from severe illness, particularly those near the end of life, aimed at improving the quality of life for patients, their families, and caregivers [1]. Palliative care service is not limited to end-of-life scenarios but applies to anyone experiencing profound health-related suffering, regardless of age or disease stage [2]. The approach includes prevention, early identification, comprehensive assessment, and management of physical, psychological, social, and spiritual issues [1].

Globally, over 56.8 million people require palliative care each year, yet only a small proportion about 14% receive it, with access particularly limited in low- and middle-income countries (LMICs) where 78% of those in need reside [3].

Many countries still lack national policies, essential medications, trained healthcare professionals, and community-based care models for palliative care [4–6]. These systemic gaps result in inadequate service provision, leaving millions of patients with serious illnesses to suffer from unmanaged pain, psychological distress, and poor quality of life. The global consequences are severe, millions die each year in avoidable suffering, reflecting a profound failure of health systems and raising significant ethical and human rights concerns [3,4,6]. This crisis highlights the urgent need to integrate palliative care into universal health coverage and to strengthen healthcare systems with culturally appropriate and sustainable approaches [7].

In LMICs, such as Ethiopia, palliative care utilization is hindered by a lack of trained healthcare professionals, poor integration into health systems, and insufficient access to essential medications [6,8,9]. Cultural misconceptions, poor patient awareness, and social stigma further impede uptake [4]. Structural issues like inadequate funding, geographic inaccessibility, and lack of national policies exacerbate the issue of palliative care, limiting access and contributing to widespread suffering and poor quality of life [10].

Ethiopia's palliative care services are primarily urban, underserved, and weak, with challenges like limited workforce capacity, drug availability, and cultural taboos requiring urgent policy prioritization [8,11–15].

This qualitative systematic review aims to explore the barriers and facilitators to palliative care utilization in Ethiopia by synthesizing findings from multiple primary studies. By integrating the evidence, the review seeks to provide a comprehensive understanding of the factors contributing to the underutilization of palliative care in this population and highlight facilitators. By identifying and connecting these factors across studies, the findings of this review are expected to guide future policy development, inform targeted interventions, and promote the integration of palliative care into Ethiopia's national healthcare framework, in alignment with global palliative care priorities and World Health Organization (WHO) recommendations.

## Methods

### Study design

This qualitative systematic review synthesized existing research on the barriers and facilitators to palliative care utilization in Ethiopia. The review followed the Preferred Reporting Items for Systematic Reviews and Meta-Analyses (PRISMA) 2020 [16]and the PRISMA Extension for Qualitative Evidence Synthesis (PRISMA-QS) [17] guidelines to ensure transparency and rigor. Although PRISMA-QS is still under development, it provided a useful framework for reporting qualitative synthesis.

### Aim of the study

This review aimed to provide a contextualized understanding of the factors affecting access to and use of palliative care in the Ethiopian healthcare system by synthesizing qualitative findings from primary studies.

### Review protocol and PROSPERO registration

A review protocol was developed in advance based on the Joanna Briggs Institute (JBI) Manual [18] for Evidence Synthesis to ensure methodological rigor and transparency. The protocol detailed the review objectives, eligibility criteria, search strategy, data extraction, quality appraisal tools, and synthesis methods. Although the PROSPERO registration (CRD420251027739) was completed on April 6, 2025—shortly after the initial data search and screening had begun—this was due to the necessary time to finalize the protocol to meet PROSPERO's submission requirements. All review procedures were pre-specified in the protocol and strictly followed throughout the study to minimize bias and ensure reproducibility.

### Search strategy

A comprehensive and systematic search was conducted across five electronic databases: PubMed/MEDLINE, Scopus, Web of Science, CINAHL, and Google Scholar, covering all articles published up to April 10, 2025. The search strategy included both free-text keywords and controlled vocabulary terms (e.g., MeSH), combined using Boolean operators ("Palliative Care" OR "End-of-life care" OR "Hospice") AND ("Barriers" OR "Facilitators" OR "Challenges" OR "Enablers") AND ("Qualitative") AND ("Ethiopia"). The search focused on studies addressing barriers and facilitators to integrating palliative care in Ethiopia using qualitative designs. The reference lists of all included studies were manually screened to identify additional eligible articles. The full electronic search strategies for each database are provided in the S1 Appendix (S1_Appendix.docx).

### Inclusion and exclusion criteria

Eligibility criteria were structured using the SPIDER tool, which is particularly appropriate for identifying studies in qualitative syntheses [19]. The sample (S) included patients, caregivers, and healthcare professionals involved in palliative

care delivery in Ethiopia. The phenomenon of interest (PI) focused on the utilization of palliative care services, with an emphasis on identifying both barriers and facilitators. Eligible studies employed qualitative designs (D), including in-depth interviews, focus groups, ethnographies, or other narrative approaches. Evaluation (E) centered on participants' lived experiences, perceptions, and meanings surrounding access to palliative care. Only primary research articles that were qualitative or mixed-methods with clearly separable qualitative findings (R) were included. Studies were eligible if they were conducted in Ethiopia, and focused on palliative care. The search was not restricted by date or language to enhance inclusivity and reduce the risk of missing relevant qualitative studies. Excluded were purely quantitative studies, reviews, opinion pieces, editorials, conference abstracts, and any research conducted outside Ethiopia.

## Search and screening

Retrieved articles were imported into Zotero for reference management and duplicate removal. Two reviewers independently screened titles and abstracts, followed by full-text reviews of potentially eligible studies. Discrepancies were resolved by discussion. A PRISMA 2020 flow diagram illustrates the selection process (Fig 1).

The database was searched for every article published on palliative care services till March 10, 2025, and continued to update until we sent it for publication. The data was extracted from March 11–20 and later analyzed from March 21–30, and the report generation till April 10, 2025.

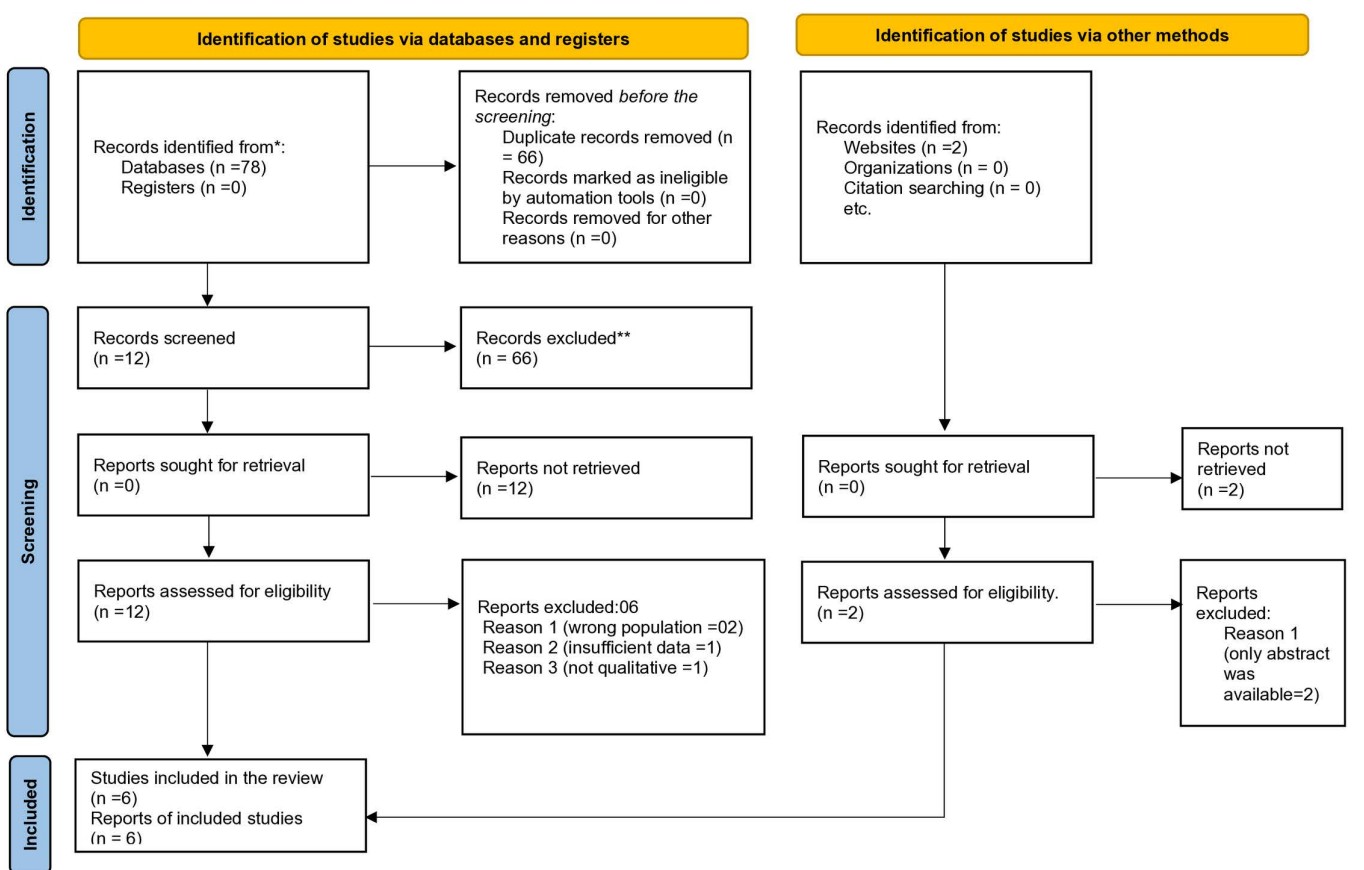

**Fig 1. PRISMA 2020 flow diagram of study selection process.**

### Quality appraisal

The methodological quality of the included studies was assessed using the Critical Appraisal Skills Programme (CASP) checklist [20], which evaluates ten domains such as research aims, data collection, reflexivity, and ethical considerations. Two reviewers independently appraised the quality of each study and recorded their judgments. While no study was excluded based solely on quality, the appraisal outcomes were used to interpret the strength and trustworthiness of the synthesized findings. Although reflexivity was inconsistently reported, we acknowledge its importance. In our synthesis, we reflected on how our professional roles may influence interpretation and theme development.

### Data extraction

A standardized, piloted form was used to extract relevant information, including study characteristics, participant details, data collection and analysis methods, key themes, and participant quotes. Two reviewers performed the extraction independently, and discrepancies were resolved through dialogue and consensus, ensuring the reliability and accuracy of the data extracted for synthesis.

### Data completeness

All included studies provided sufficient qualitative data relevant to the review objectives. There were no notable gaps in reporting that affected the synthesis or interpretation of findings. The complete data extraction table is provided in the supplementary materials for transparency (Table 1).

### Data synthesis

Thematic synthesis, following the approach by Thomas and Harden [21], was used to integrate findings. This involved line-by-line coding of study results, development of descriptive themes, and generation of analytical themes. The synthesis was iterative and collaborative, with co-authors engaging in critical reflection to refine themes, enhance credibility, and ensure resonance with the Ethiopian sociocultural and healthcare context.

### Assessment of confidence in the evidence

The Grading of Recommendations Assessment, Development and Evaluation Confidence in the Evidence from Reviews of Qualitative research(GRADE-CERQual approach) [22] was used to assess the confidence in each synthesized theme. Each key finding was assessed across four domains: methodological limitations of the contributing studies, coherence of the data supporting the finding, adequacy of the data in terms of richness and quantity, and relevance to the review question. Themes were rated as having high, moderate, low, or very low confidence, and rationales for judgments were documented.

### Ethical consideration

This study involved a secondary analysis of previously published research and did not require formal ethical approval. All efforts were made to ensure ethical integrity by including only studies that had obtained ethical clearance and informed consent from participants. Intellectual property was respected through appropriate citation and acknowledgment of original authors.

## Results

### Study selection

A total of 78 unique records were identified through database and manual searches. After screening titles and abstracts, 12 full-text articles were assessed, and 6 met the inclusion criteria. The PRISMA 2020 flow diagram (Fig 1). (Fig 1. PRISMA 2020 flow diagram of study selection process.) presents the detailed screening process.

**Table 1. Detailed table listing every study identified during our literature search, including those excluded from your analysis.**

| Study # | Citation | Title | Included? | Reason for Inclusion or Exclusion (if excluded) |
|---|---|---|---|---|
| 1 | **Abate et al. BMC Palliative Care (2023) 22:57,** https://doi.org/10.1186/s12904-023-01181-w | Barrier analysis for continuity of palliative care from health facility to household among adult cancer patients in Addis Ababa, Ethiopia | Yes | This study used a qualitative approach to explore palliative care access and utilization among patients, in Ethiopia. The thematic analysis directly addressed key barriers and facilitators to palliative care use, making it highly relevant to the review objectives. |
| 2 | **Aregay et al. BMC Palliative Care (2023) 22:156** https://doi.org/10.1186/s12904-023-01283-5 | Palliative care in Ethiopia's rural ®ional health care settings: a qualitative study enabling factors& implementation challenges | Yes | This study explores palliative care access and utilization among patients, in Ethiopia. The thematic analysis directly addressed key barriers and facilitators to palliative care use, making it highly relevant to the review objectives. |
| 3 | Atsede Aregay, Margaret O'Connor, Jill Stow, Nicola Ayers, and Susan Lee,2023 https://doi.org/10.1177/26323524231198542 https://doi.org/10.1177/26323524231198542 | Perceived policy-related barriers to palliative care implementation: a qualitative study | Yes | This study used a qualitative approach to explore palliative care access and utilization among patients, in Ethiopia. |
| 4 | **Aregay A, O'Connor M, Stow J, Ayers N, Lee S (2024) Measuring and exploring the barriers to translating palliative care knowledge into clinical practice in rural and regional health-care settings. Palliative and Supportive Care 22(6), 1605–1614.** https://doi.org/10.1017/S1478951523000755 | Measuring and exploring the barriers to translating palliative care knowledge into clinical practice in rural and regional health-care settings | Yes | The thematic analysis directly addressed key barriers and facilitators to palliative care use, making it highly relevant to the review objectives. |
| 5 | Kaba M, de Fouw M, Deribe KS, Abathun E, Peters AAW, Beltman JJ (2021) Palliative care needs and preferences of female patients and their caregivers in Ethiopia: A rapid program evaluation in Addis Ababa and Sidama zone. PLoS ONE 16(4): e0248738. https://doi.org/10.1371/journal.pone.0248738 | Palliative care needs and preferences of female patients and their caregivers in Ethiopia: A rapid program evaluation in Addis AbabaandSidamazone | Yes | This study used a qualitative approach to explore palliative care access and utilization among patients, in Ethiopia. The thematic analysis directly addressed key barriers and facilitators to palliative care use. |
| 6 | Endalew Hailu Negasa, Sarie Petronella Human & Ameyu Godesso Roro To cite this article: Endalew Hailu Negasa, Sarie Petronella Human & Ameyu Godesso Roro (2023) Challenges in Palliative Care Provision in Ethiopia: An Exploratory Qualitative Study, Journal of Pain Research, 3405–3415, https://doi.org/10.2147/JPR.S415866 To link to this article: https://doi.org/10.2147/JPR.S415866 | Challenges in Palliative Care Provision in Ethiopia: An Exploratory Qualitative Study | Yes | This study used a qualitative approach to explore palliative care access and utilization in Ethiopia. |
| 7 | Eleanor Anderson Reid, MD, MSc, DTM&H,1 Esayas Kebede Gudina, MD, DTM&H, PhD,2 Nicola Ayers, PhD, MSc, BSc (Hons), RGN,3,4 Wondimagegnu Tigineh, MD,5 and Yoseph Mamo Azmera, JOURNAL OF PALLIATIVE MEDICINE Volume 21, Number 5, 2018 a Mary Ann Liebert, Inc. https://doi.org/10.1089/jpm.2017.0419 | Caring for Life-Limiting Illness in Ethiopia: A Mixed-Methods Assessment of Outpatient Palliative Care Needs | No | The study by Reid et al. (2025) employed a mixed-methods design with a primary focus on assessing the overall burden, costs, and symptomatology associated with life-limiting illness, rather than exploring barriers and facilitators to palliative care utilization. While qualitative data were collected, the study does not offer in-depth thematic analysis specifically targeting the utilization of palliative care services and therefore does not meet the inclusion criteria for a qualitative systematic review focused on utilization-related factors. |

*(Continued)*

**Table 1.** (Continued)

| Study # | Citation | Title | Included? | Reason for Inclusion or Exclusion (if excluded) |
|---|---|---|---|---|
| 8 | Atalay Mulu Fentie, Anteneh Belete & Muluken Nigatu Selam To cite this article: Atalay Mulu Fentie, Anteneh Belete & Muluken Nigatu Selam (2023) Challenges of Access to Oral Morphine Medicine: Palliative Care at a Crossroads for Cancer Patients in Ethiopia, Journal of Pain Research, 1829–1833, https://doi.org/10.2147/JPR.S410944 To link to this article: https://doi.org/10.2147/JPR.S410944 | Challenges of Access to Oral Morphine Medicine: Palliative Care at a Crossroads for Cancer Patients in Ethiopia | No | This study focuses primarily on challenges related to the availability and access to oral morphine for pain management, rather than exploring broader barriers and facilitators to palliative care utilization through qualitative methods. Therefore, it does not meet the inclusion criteria for a qualitative systematic review. |
| 9 | Reid EA, Abathun E, Diribi J, et al. BMJ Supportive & Palliative Care Epub ahead of print: [please include Day Month Year]. https://doi.org/10.1136/spare-2022–003996 | Early palliative care in newly diagnosed cancer in Ethiopia: feasibility randomized controlled trial and cost analysis | No | This study primarily reports on the feasibility and outcomes of a randomized controlled trial assessing early palliative care interventions, with a focus on clinical effectiveness and cost analysis. It does not employ qualitative methodology aimed at exploring barriers or facilitators to palliative care utilization and therefore does not meet the inclusion criteria for a qualitative systematic review. |
| 10 | Kaba M, de Fouw M, Deribe KS, Abathun E, Peters AAW, Beltman JJ (2021) Palliative care needs and preferences of female patients and their caregivers in Ethiopia: A rapid program evaluation in Addis Ababa and Sidama zone. PLoS ONE 16(4): e0248738. https://doi.org/10.1371/journal.pone.0248738 | Palliative care needs and preferences of female patients and their caregivers in Ethiopia: A rapid program evaluation in Addis Ababa and Sidama zone | No | This study's primary focus was on evaluating existing program services rather than specifically identifying barriers and facilitators to palliative care utilization. As such, it does not fully meet the inclusion criteria for a qualitative systematic review centered on utilization factors. |
| 11 | Muday Beneberu1, Getachew Teshale1,2*, Kaleb Assegid Demissie1, Endalkachew Dellie1, Melak Jejaw1 and Asmamaw Atnafu, BMC Palliative Care (2025) 24:57 https://doi.org/10.1186/s12904-025-01694-6 | Patient-centeredness and determinant factors of palliative care service for adult cancer patients in public hospitals of Addis Ababa, Ethiopia, 2024: cross-sectional mixed method study | No | This study used a mixed-methods design primarily focused on measuring the level and determinants of patient-centeredness in palliative care services, rather than exploring barriers and facilitators to palliative care utilization through a qualitative lens. Therefore, it does include criteria for a qualitative systematic review. |
| 12 | Yoseph Mamo, Anteneh Habte, Nardos W/Giorgis1, Aynalem Abreha3, Nicola Ayers4, Ephrem Abathun1, Eleanor Reid5, Mirgissa Kaba Ethiop. J. Health Dev. 2020; 34(4):310–312] https://www.ajol.info/index.php/ejhd/article/view/203468/191888 | The evolution of hospice and palliative care in Ethiopia: From historic milestones to future directions | No | This article is a narrative review that outlines the historical development and future directions of hospice and palliative care in Ethiopia. It does not present original qualitative data focused on barriers and facilitators to palliative care utilization, and therefore does not meet the inclusion criteria for a qualitative systematic review. |

The six included studies were conducted in various settings across Ethiopia, providing insights from both rural and urban contexts [9,14,23–25]. Abate et al. [23] focused on continuity of care between health facilities and households (Table 2).

Aregay et al. (2023a) [25] and Aregay et al. (2024) [26] examined service delivery in rural and regional hospitals. Aregay et al. (2023b) [25] explored policy-level barriers, while Kaba et al. [14] evaluated urban palliative care programs. Negasa et al. [9]examined systemic challenges from both central and regional health institutions.

All the studies employed qualitative designs [9,14,23–25], except Aregay et al.[26], which adopted a mixed-methods approach with a delineated qualitative component. Interviews were the primary data collection method in all studies [9,14,23–26], supported by thematic analysis using tools such as ATLAS.ti by Abate et al. [23] and Negasa et al.[9], and NVivo was used in Aregay et al.[26]. Methodological approaches ranged from rapid evaluations to key informant

**Table 2. Characteristic of included study on palliative care in Ethiopia, 2025.**

| Author(s) & Year | Study Setting | Research Objectives | Study Design | Participant Characteristics | Data Collection & Analysis Methods | Key Findings (Barriers) & Illustrative Participant Quotes | Key Findings (Facilitators) & Illustrative Participant Quotes | Data Extractor(s) | Name of Data Collector |
|---|---|---|---|---|---|---|---|---|---|
| Abate et al., 2023 | Health facilities and households, in Ethiopia | To explore barriers to continuity of palliative care from health facility to household | Qualitative study | 19 participants - Sex: 10 female, 9 male, Age: 25–60 years. Residence: Urban (mostly) Roles: Healthcare providers, caregivers, health extension workers | In-depth interviews; thematic analysis by ATLAS.ti | Lack of clear national policy *"Palliative care is not included in performance plans, so staff don't consider it a priority."* Poor resource allocation *"There is no budget line for palliative care in the ministry's structure."* Limited stakeholder engagement (Community-Level Neglect) *"The policy doesn't address palliative care at the community level."* Leadership and Implementation Issues: *"We lack leadership commitment and clear guidelines for implementation."* | Family support, community-based services. *"Community support helps us a lot, especially when professionals don't show up."* | March 11/20/2027 | Sadik Abdulwehab and Frezer Kedir |
| Aregay et al., 2023a | Rural and regional healthcare settings in Ethiopia | To explore enabling factors and challenges in delivering palliative care | Qualitative study | 29 participants - Roles: Healthcare providers, facility managers - Setting: Rural and regional healthcare facilities | In-depth interviews; thematic analysis | Low knowledge, high workload, and poor integration of palliative care into the health system. Lack of Dedicated Teams: *"We don't have a dedicated palliative care team in our facility."* Unstructured Service Provision: *"There is no organized palliative care service; we just do what we can."* Centralized Services and Rural Burden: *"In rural areas, the families suffer most because services are centralized."* Overburdened and Undertrained Staff: *"We are overloaded and lack training, but we try to help when possible."* | Staff motivation, community involvement, and leadership support. - *"What keeps us going is our motivation and the support of the community."* | March 12/20/2027 | Sadik Abdulwehab and Frezer Kedir |
| Aregay et al., 2023b | Various Ethiopian regions, policy context | To identify policy-related barriers to palliative care implementation | Qualitative descriptive study | Total: 25 participants - Roles: Policymakers, program directors, healthcare administrators - Setting: Regional and national health offices | Key informant interviews; thematic analysis | Lack of clear national policy, poor resource allocation, and limited stakeholder engagement. Policy Exclusion from Strategic Plans: *"Palliative care is not included in performance plans, so staff don't consider it a priority."* Absence of Dedicated Budget: *"There is no budget line for palliative care in the ministry's structure."* Neglect of Community-Level Integration: *"The policy doesn't address palliative care at the community level."* Weak Leadership and Implementation Frameworks: *"We lack leadership commitment and clear guidelines for implementation."* | Stakeholder Awareness and Support - *"We need stakeholders who understand the importance of palliative care."* | March 12/20/2027 | Sadik Abdulwehab and Frezer Kedir |

*(Continued)*

**Table 2.** (Continued)

| Author(s) & Year | Study Setting | Research Objectives | Study Design | Participant Character-istics | Data Collec-tion & Analysis Methods | Key Findings (Barriers) & Illustrative Participant Quotes | Key Findings (Facilitators) & Illustrative Participant Quotes | Data Extractor(s) | Name of Data Col-lector |
|---|---|---|---|---|---|---|---|---|---|
| Aregay et al., 2024 | One region in Ethiopia with comprehensive, general, and primary hospitals | To measure and explore the barriers to translating theoretical knowledge of palliative care into clinical practice | Mixed-method (cross-sectional survey + qualitative interviews) | 173 nurses for the survey; 42 professionals for interviews including nurses, doctors, pharmacists, policymakers | Survey tools (PCQN, FAT-COD, practice scale); thematic analysis using NVivo | Knowledge Deficits: "We do not have detailed information on how to provide care for chronically ill patients. We need guidance on providing palliative care." Access to Guidelines: "We do not know about the documents [palliative care] … we do not have updated documents in this institute." Curriculum Gaps: "There is no dedicated chapter or topic on palliative care in the under-graduate program." Inadequate Training Methods: "The traditional lecture-based system… does not work. We have to follow a different system… such as training with clinical placement. Systemic Constraints: Shortages in medicine, staff, and financial resources; weak policy emphasis. | Positive Attitudes Toward Palliative Care: "We know that it [palliative care] starts from diagnosis … until the end of life Inclusion in Some Curricula: Certain post-graduate and diploma programs include palliative care content. Recognition of Need for Policy Engagement: Highlighted the need for changing teaching methods and involving policymakers. | March 13–16/20/2027 | Sadik Abdul-wehab and Frezer Kedir |
| Kaba et al., 2021 | Addis Ababa and Yirgalem, Ethiopia | To explore palliative care needs and preferences of female patients and caregivers, and perspectives of stakeholders on service provision | Rapid program evaluation (qualitative) | 77 interviews (34 patients, 12 primary caregivers, 15 voluntary caregivers, 16 stakeholders) | In-depth interviews; inductive thematic analysis | Limited Awareness Among Patients & Caregivers: "Most patients and caregivers reported that they 'never heard' of palliative care and 'don't understand' what palliative care is." Insufficient Psychosocial & Economic Support: Services mainly addressed pain but lacked emotional, spiritual, and economic support. Weak Referral Systems: Only Hospice Ethiopia had formal referral pathways. Lack of Trained Providers in Some Programs: MJDA and B4G lacked formally trained palliative care professionals. | Effective Holistic Care Models (Hospice Ethiopia): Provided medical, psychosocial, financial, and spiritual support, including home-based and day-care. Community Engagement: The involvement of bidders, religious leaders, and community volunteers enhanced care. Recognition of Family and Community Role: Strong informal care systems and religious support networks. | March 17–18/20/2027 | Sadik Abdul-wehab and Frezer Kedir |

*(Continued)*

**Table 2.** (Continued)

| Author(s) & Year | Study Setting | Research Objectives | Study Design | Participant Characteristics | Data Collection & Analysis Methods | Key Findings (Barriers) & Illustrative Participant Quotes | Key Findings (Facilitators) & Illustrative Participant Quotes | Data Extractor(s) | Name of Data Collector |
|---|---|---|---|---|---|---|---|---|---|
| Negasa et al., 2023 | Addis Ababa and Jimma Zone | To examine the challenges of palliative care provision in Ethiopia | Exploratory qualitative study | 29 key informants and 5 FGDs with nurses in chronic care clinics | Thematic analysis using ATLAS-ti | Patient-Related Challenges: *Delay in care-seeking, discontinuation due to cost, cultural preference for dying at home.* Provider-Related Issues: *"Lack of awareness of palliative care," "no training," "absent in curricula."* Health System Gaps: *Lack of medications, chemotherapy, radiotherapy; poor facility-community linkage.* Weak Collaboration: *"Limited partnership between government and NGOs impedes service integration."* | Stakeholder Willingness: *Health professionals and community leaders acknowledge the importance of palliative care.* Existing NGO and Faith-Based Organization Involvement: *Some local and faith-based groups provide care and support at the grassroots level.* Community Support: *Traditional care and volunteerism were active in some settings.* | March 20/20/2027 | Sadik Abdulwehab and Frezer Kedir |

interviews [9,14,23–25]. Participants included healthcare providers, policy actors, caregivers, and patients [9,14,23–26]. This diverse representation enabled a multidimensional understanding of palliative care delivery challenges and enablers in Ethiopia.

## Methodological quality of included studies

The methodological quality of the included studies was appraised using the CASP checklist. All studies had a clear statement of research aims, employed appropriate qualitative methodologies, and utilized research designs aligned with their objectives. Most studies (83%) applied sound recruitment strategies and effective data collection methods, and all conducted rigorous data analysis [9,14,23–25]. However, only two studies [14,26] adequately discussed reflexivity, indicating a common limitation in reporting researcher influence (Table 3).

## Thematic synthesis: Barriers and facilitators affecting palliative care delivery in Ethiopia

The qualitative synthesis of the included studies revealed two overarching categories: barriers and facilitators to the implementation and delivery of palliative care in Ethiopia. Within the barriers category, five major themes emerged: (1) policy and governance gaps, (2) health system challenges, (3) knowledge and training deficits, (4) sociocultural and economic barriers, and (5) collaboration challenges. These themes reflect persistent structural, educational, and systemic obstacles

**Table 3. Critical appraisal skills programme checklist for qualitative research quality appraisal.**

| Author and year | 1. Was there a clear statement of the aims of the research? | 2. Is a qualitative methodology appropriate? | 3. Was the research design appropriate to address the aims of the research? | 4. Was the recruitment strategy appropriate to the aims of the research? | 5. Was the data collected in a way that addressed the research issue? | 6. Has the relationship between the researcher and participants been adequately considered? | 7. Have ethical issues been taken into consideration? | 8. Was the data analysis sufficiently rigorous? | 9. Is there a clear statement of findings? | 10. How valuable is the research? |
|---|---|---|---|---|---|---|---|---|---|---|
| Abate et al., 2023 | Yes | Yes | Yes | Yes | Yes | Partially considered | Not stated | Yes | Yes | High |
| Aregay et al., 2023a | Yes | Yes | Yes | Yes | Yes | Partially considered | Not stated | Yes | Yes | High |
| Aregay et al., 2023b | Yes | Yes | Yes | Yes | Yes | Not stated | Not stated | Yes | Yes | High |
| Aregay et al., 2024 | Yes | Yes | Yes | Yes | Yes | Partially considered | Not stated | Yes | Yes | High |
| Kaba et al., 2021 | Yes | Yes | Yes | Yes | Yes | Partially considered | Yes | Yes | Yes | High |
| Negasa et al., 2023 | Yes | Yes | Yes | Yes | Yes | Unclear/No | Not stated | Yes | Yes | High |

that impede the provision of effective palliative care. Illustrative quotes highlight issues such as the absence of national policies, limited professional training, inadequate infrastructure, cultural preferences for dying at home, and lack of coordinated stakeholder efforts (Table 4).

Conversely, five key themes were identified under facilitators: (1) community and family support, (2) healthcare provider motivation, (3) inclusion of palliative care in education and training curricula, (4) adoption of holistic care models, and (5) active involvement of stakeholders. These enablers emphasize the importance of grassroots support, the dedication of healthcare workers, the integration of palliative care into academic programs, and the collaborative roles of NGOs and faith-based organizations. A summary of these themes, illustrative quotations, and contributing studies is presented in Table (Table 5).

### Theme formulated as barriers to palliative care

**Policy and governance gaps.** The theme of policy and governance gaps emerged prominently across the included studies and consisted of four interrelated subthemes: absence of national policy, limited stakeholder engagement, lack of a dedicated budget, and weak leadership and implementation frameworks. The absence of an articulated national policy for palliative care was cited in multiple studies [23,25]. This policy void was seen as a foundational barrier, undermining the institutional prioritization of palliative care. In several cases, respondents emphasized that palliative care was not integrated into national or regional performance frameworks, reducing its visibility within public health agendas: "Palliative care is not included in performance plans, so staff don't consider it a priority" [23].

Stakeholder engagement was also found to be minimal, particularly at the community level, where policies often failed to address care continuity from health facilities to homes. Aregay et al. [25] highlighted that the absence of community representation in strategic planning led to neglect in rural and underserved areas. Budget limitations were closely linked to this strategic neglect, with multiple studies noting the absence of a defined financial line for palliative care services within governmental structures: "There is no budget line for palliative care in the ministry's structure" [23]. This fiscal exclusion

**Table 4. Selected quotes from studies on barriers to palliative care: Themes, subthemes, and illustrative quotes.**

| Theme | Subtheme | Illustrative Quote | Contributing Studies |
|---|---|---|---|
| **Policy and Governance Gaps** | Lack of national policy | "Palliative care is not included in performance plans..." | Abate et al., 2023; Aregay et al., 2023b |
| | Limited stakeholder engagement | "The policy doesn't address palliative care at the community level." | Abate et al., 2023; Aregay et al., 2023b |
| | The absence of a dedicated budget | "There is no budget line for palliative care in the ministry's structure." | Abate et al., 2023; Aregay et al., 2023b |
| | Weak leadership and planning | "We lack leadership commitment and clear guidelines for implementation." | Abate et al., 2023; Aregay et al., 2023b |
| **Health System Challenges** | Lack of trained staff & overburden | "We are overloaded and lack training, but we try to help when possible." | Aregay et al., 2023a |
| | Poor service integration | "There is no organized palliative care service; we just do what we can." | Aregay et al., 2023a |
| | Weak referral systems | "Only Hospice Ethiopia had formal referral pathways." | Kaba et al., 2021 |
| | Rural service inaccessibility | "In rural areas, the families suffer most because services are centralized." | Aregay et al., 2023a |
| | Shortage of medicines and equipment | "There are no medicines, chemotherapy, or radiotherapy." | Negasa et al., 2023 |
| **Knowledge and Training Gaps** | Lack of provider knowledge | "We do not have detailed information on how to provide care…" | Aregay et al., 2024 |
| | No training in curricula | "There is no dedicated chapter or topic on palliative care in the undergraduate program." | Aregay et al., 2024; Negasa et al., 2023 |
| | Inadequate training approaches | "The traditional lecture-based system... does not work." | Aregay et al., 2024 |
| **Sociocultural and Economic** | Cultural preference for dying at home | "Some patients prefer to die at home... believing it's more respectful." | Negasa et al., 2023 |
| | Financial constraints | "Discontinuation of care due to cost is common." | Negasa et al., 2023 |
| | Lack of emotional and spiritual support | "We address pain, but there's no spiritual or economic support." | Kaba et al., 2021 |
| **Collaboration Challenges** | Weak NGO–government coordination | "Limited partnership between government and NGOs impedes service integration." | Negasa et al., 2023 |

perpetuated resource constraints and further discouraged the formal development of service models. Lastly, leadership and implementation barriers were widely reported. A consistent concern across studies was the lack of committed leadership and clear implementation frameworks to support even the limited palliative care services that exist [25]. Collectively, these subthemes reflect a broader systemic marginalization of palliative care within the health governance landscape in Ethiopia, contributing to fragmented service delivery and inequitable access.

**Health system challenges.** Challenges included staff shortages, poor service integration, urban-rural disparities, weak referral systems, and insufficient infrastructure. One participant noted: "We are overloaded and lack training, but we try to help when possible" [24]. Another added: "There is no organized palliative care service; we just do what we can" [24]. Furthermore, shortages in medicines and medical infrastructure were identified as ongoing challenges [9].

**Knowledge and training deficits.** This theme included limited provider awareness, absence of palliative care in educational curricula, and inadequate training methods. Healthcare professionals expressed uncertainty in delivering care due to insufficient knowledge: "We do not have detailed information on how to provide care for chronically ill patients" [26]. Participants also indicated that training programs did not adequately prepare them for real-world scenarios: "The traditional lecture-based system… does not work" [26]. The lack of curriculum content in undergraduate programs further contributes to this knowledge gap [9].

**Sociocultural and economic barriers.** This theme encompassed cultural preferences, financial hardship, and inadequate psychosocial and spiritual support. Some participants highlighted a cultural preference for dying at home,

**Table 5. Selected quotes from studies on facilitators to palliative care: Themes, subthemes, and illustrative quotes.**

| Theme | Subtheme | Illustrative Quote | Contributing Studies |
|---|---|---|---|
| Community and Family Support | Community-based care involvement | "Community support helps us a lot, especially when professionals don't show up." | Abate et al., 2023 |
| | Family and religious care networks | "We work with bidders, church leaders, and neighbors to support patients." | Kaba et al., 2021 |
| | Grassroots volunteerism | "Traditional care and volunteerism were active in some settings." | Negasa et al., 2023 |
| Health-care Staff Motivation | Personal commitment to care | "What keeps us going is our motivation and the support of the community." | Aregay et al., 2023a |
| | Positive perception of palliative care | "We know that it [palliative care] starts from diagnosis … until the end of life." | Aregay et al., 2024 |
| Education and Curriculum | Postgraduate program inclusion | "Certain postgraduate and diploma programs include palliative care content." | Aregay et al., 2024 |
| | Calls for educational reform | "We need to change how we teach this — involve clinical placements and policy." | Aregay et al., 2024 |
| Holistic Models of Care | Multidimensional services at specialty sites | "Hospice Ethiopia provided medical, psychosocial, financial, and spiritual support." | Kaba et al., 2021 |
| Stakeholder Involvement | NGO and faith-based service delivery | "Some local and faith-based groups provide care and support at the grass-roots level." | Negasa et al., 2023 |
| | Growing policy awareness | "We need stakeholders who understand the importance of palliative care." | Aregay et al., 2023b; Negasa et al., 2023 |

which limits formal care utilization: "Patients prefer to die at home... believing it's more respectful" [9]. Financial barriers were equally pronounced, with many families unable to sustain care: "Discontinuation of care due to cost is common" [9]. Emotional and spiritual needs were frequently unmet: "We address pain, but there's no spiritual or economic support" [14].

**Collaboration challenges.** Weak coordination between government and non-governmental actors was a recurrent issue. For instance, limited partnerships between the public health sector and NGOs were found to hinder service integration: "Limited partnership between government and NGOs impedes service integration" [9]. This lack of coordinated effort reduces resource pooling, knowledge exchange, and scalability of care models.

### Theme formulated as facilitators to palliative care

**Community and family support.** This facilitating theme included community-based involvement, religious networks, and informal caregiving systems. Participants frequently credited community support for sustaining care delivery in the absence of formal mechanisms: "Community support helps us a lot, especially when professionals don't show up" [23]. Religious and family structures often acted as informal care systems, particularly in rural and underserved settings [9,14].

**Healthcare provider motivation.** Healthcare staff's intrinsic motivation and professional commitment were repeatedly highlighted as critical facilitators. Despite limited resources, providers expressed a sense of duty and perseverance: "What keeps us going is our motivation and the support of the community" [24]. This motivation played a crucial role in maintaining service continuity under challenging conditions.

**Education and curriculum inclusion.** Positive developments in integrating palliative care into postgraduate and diploma-level curricula were noted. Certain academic programs now include palliative care components, providing a foundation for future capacity building: "Certain postgraduate and diploma programs include palliative care content" [26]. Participants also called for reforming teaching methods to better match real-world practice.

**Holistic care models.** Specialized centers such as Hospice Ethiopia were cited for offering comprehensive, patient-centered care models that integrate medical, psychosocial, and spiritual services: "Hospice Ethiopia provided medical, psychosocial, financial, and spiritual support" [14]. These models serve as benchmarks for scaling up integrated palliative care.

**Stakeholder involvement.** Stakeholder recognition of the importance of palliative care and growing engagement from NGOs and religious institutions were seen as facilitators. "Some local and faith-based groups provide care and support at the grassroots level," noted one participant [9]. Additionally, there is an increasing policy momentum: "We need stakeholders who understand the importance of palliative care" [25].

## Confidence in review findings (GRADE-CERQual assessment)

To assess the trustworthiness of each synthesized theme, the GRADE-CERQual approach was applied across four domains: methodological limitations, data coherence, adequacy, and relevance to the review question. Each theme was assigned a confidence level of high, moderate, low, or very low based on these criteria, with justifications documented to ensure transparency (Table 6).

**Table 6. Updated GRADE-CERQual summary of qualitative findings.**

| Theme | Methodological Limitations | Coherence | Adequacy | Relevance | Confidence Level | Rationale |
|---|---|---|---|---|---|---|
| Policy and Governance Gaps | Low: Studies used appropriate qualitative approaches with clear policy-focused participant roles. | High: Multiple studies agree on the lack of clear national policy and leadership. | Moderate: Sufficient quotes from policy-level informants. | High: Directly relevant to palliative care implementation. | Moderate | Strong coherence but limited participant diversity at higher policy levels. |
| Health System Challenges | Low: Consistently strong designs and data collection methods. | High: Clear agreement on resource shortages and unstructured care systems. | High: Detailed descriptions and rich quotes from multiple roles. | High: Well aligned with the health system context in Ethiopia. | High | Robust evidence across regions and professions. |
| Knowledge and Training Gaps | Low to Moderate: Some reliance on mixed-methods data. | Moderate: Consistent but varied interpretation of training gaps. | Moderate: Adequate range of participants; some gaps in student perspectives. | High: Training directly affects care provision. | Moderate | Sound evidence base with minor coherence variation. |
| Sociocultural and Economic Factors | Low: Clear qualitative approaches from multiple studies. | High: Recurrent themes of cultural norms and financial hardship. | High: Strong patient and caregiver voices represented. | High: Essential to understand barriers to care-seeking& service use. | High | Rich, relevant data with minimal limitations. |
| Collaboration Challenges | Low: Well-structured interviews with relevant stakeholders. | Moderate: Some variation in views on inter-agency collaboration. | Moderate: Adequate quotes from NGO/government staff; fewer from patients. | High: Directly relates to service integration. | Moderate | Evidence is solid but stakeholder representation is slightly uneven. |
| Community and Family Support | Low: Community-focused studies well-executed. | High: Strong alignment across diverse regions. | High: Rich descriptions of informal care systems. | High: Central to culturally appropriate care models. | High | Clear and consistently reported facilitator. |
| Healthcare Staff Motivation | Low: Quotes and observations from multiple staff roles. | High: Motivation and commitment appear across studies. | Moderate: Few direct quotes but strong narrative presence. | High: Relevant to implementation sustainability. | Moderate | Well-supported but could benefit from deeper exploration. |
| Education and Curriculum Improvements | Low: Mixed but mostly robust study designs. | Moderate: Curriculum inclusion varies across institutions. | Moderate: Some supporting quotes; limited student data. | High: Curriculum directly shapes knowledge and attitudes. | Moderate | Useful insights but not fully explored in all studies. |
| Holistic Models of Care | Low: Qualitative data from program evaluations. | High: Hospice Ethiopia is cited consistently as a positive model. | High: Multiple quotes from patients and staff. | High: Highlights alternative service models. | High | Well-documented with diverse perspectives. |
| Stakeholder Involvement | Low: Direct interviews with decision-makers and NGOs. | Moderate: Mixed views on actual vs. ideal involvement. | Moderate: Limited but meaningful quotes from stakeholders. | High: Central to systemic change. | Moderate | Important theme with moderate evidentiary support. |

Of the ten identified themes, four were rated with high confidence; Health System Challenges, Sociocultural and Economic Factors, Community and Family Support, and Holistic Models of Care as they were strongly supported by rich, consistent evidence from multiple well-conducted studies. The remaining five themes; Policy and Governance Gaps, Knowledge and Training Deficits, Collaboration Challenges, Healthcare Provider Motivation, and Stakeholder Involvement were graded as moderate confidence due to some methodological concerns or limited participant diversity. No theme was rated as low or very low confidence. This assessment reinforces the robustness of the findings while also highlighting priority areas for future research, particularly in strengthening policy frameworks, inter-organizational collaboration, and workforce training in palliative care.

## Discussion

This is the first qualitative review in Ethiopia on palliative care services that identified five major barriers to effective palliative care implementation: policy and governance gaps, health system challenges, knowledge and training deficits, sociocultural and economic barriers, and collaboration challenges. This review focuses on low-income settings in Ethiopia, where regional disparities between urban centers and rural areas complicate equitable access to palliative care.

This review found that policy and governance gaps are a major barrier to the implementation of palliative care in Ethiopia. These challenges are not unique to Ethiopia, as similar barriers have been reported across various LMICs [27,28], and Eastern Europe [29] identified the absence of comprehensive national guidelines, insufficient integration of palliative care into existing health systems, and limited financial support as key obstacles to effective palliative care delivery. These parallels underscore the widespread nature of policy and governance issues hindering palliative care across various countries.

The absence of a formal national palliative care policy in Ethiopia has been identified as a significant structural barrier to effective palliative care delivery. This issue mirrors challenges observed in other LMCs [28], where palliative care services are often not integrated into national health systems as studies done in India [30] and Egypt [31]. Findings from this study emphasize the necessity for Ethiopia to establish a comprehensive national palliative care policy, which should integrate palliative care into all healthcare systems, guarantee adequate professional training, and secure sustainable funding. Addressing these gaps is essential to improve the quality of life for patients with life-limiting illnesses and to align Ethiopia's healthcare services with global standards for palliative care.

The lack of involvement of community members, healthcare providers, and local leaders in the development of palliative care policies in Ethiopia significantly hinders their relevance and effectiveness, leading to poorly understood and inadequately implemented policies, similar to Sub-Saharan Africa's study [32]. These findings collectively highlight the critical importance of inclusive stakeholder engagement in the development and implementation of palliative care policies. Ensuring that all relevant parties are actively involved can lead to more effective, culturally appropriate, and sustainable palliative care services and improve health outcomes.

Many studies indicated that community members, healthcare providers, and local leaders are often not involved in policymaking, which limits the relevance and feasibility of the resulting strategies., This is similar to the study done in Kenya and Uganda [33]. The Ethiopian context reflects this, where stakeholders at the implementation level feel disconnected from strategic planning.

Another significant challenge identified was the lack of a dedicated budget for palliative care, even the best-designed policies remain unimplemented, which aligns with findings from the Lancet Commission on Global Access to Palliative Care and Pain Relief [6]. which highlighted that palliative care services in Africa are frequently underfunded and reliant on external donor support, jeopardizing their sustainability. Participants included in the original article of this study, particularly Ethiopian healthcare professionals, voiced similar concerns about the lack of sustainable funding and its impact on service delivery.

Weak leadership and the absence of structured implementation mechanisms were cited as barriers to the scale-up of palliative care, which is similar to studies done in Norway [34], Israel [35], United States [36]. They highlight the need for dedicated leadership, comprehensive training, and the integration of palliative care into organizational policies to overcome existing barriers and improve care delivery.

Ethiopia's health system faces challenges such as a shortage of trained staff, poor service integration, weak referral systems, urban-rural disparities, and inadequate infrastructure, which hinders palliative care delivery, affecting pain management and holistic support [9,24]. These challenges are not isolated to Ethiopia, with similar findings in other sub-Saharan African countries [28]. Addressing these barriers requires comprehensive reforms, including policy changes, improved healthcare workforce training, and better resource allocation to ensure equitable access to palliative care across all regions.

The lack of structured palliative care education in both undergraduate and in-service training programs in Ethiopia is causing significant knowledge and training deficits among healthcare professionals, leading to uncertainty in delivering comprehensive care [9,26], as highlighted in a study conducted in the country [37]. These findings highlight the urgent need for curriculum reform and standardized training to build a competent palliative care workforce.

Sociocultural and economic barriers become another theme that significantly limits the utilization of palliative care services in Ethiopia [9,14], consistent with findings from Ghana [27], Iran [38], and other LMICs [28]. In Ethiopia, cultural and religious norms significantly influence end-of-life care preferences, with many patients and families favoring home-based care due to its alignment with their spiritual values and spiritual fulfillment, emphasizing the need for culturally sensitive palliative care models and this can hinder access to formal palliative care, leading to alternative treatments and exclusion from national health insurance schemes [8,23]. These factors underscore the need for culturally sensitive, economically accessible, and integrated palliative care strategies.

Collaboration challenges between governments and non-governmental organizations significantly hinder the integration and delivery of palliative care services in Ethiopia [9], which is similar to a study done in Zimbabwe [39], India [40], and other LMICs [28]. The global expansion of palliative care services faces challenges due to policy differences and lack of integration into national health frameworks, despite successful NGO-led models, highlighting the need for stronger policy-based collaboration.

Community and Family Support themes were most commonly reported from factors facilitating palliative care service utilization in Ethiopia [9,14,23]. The study highlights the importance of community involvement, religious networks, and informal caregiving systems in sustaining care delivery, especially in rural and underserved settings, where formal mechanisms may not be available, which is a similar finding in a study done in the Netherlands [41], Colombia [42], India [43]. Strengthening community-based approaches and promoting active family participation can enhance accessibility, sustainability, and quality of services, especially in underserved areas, and should be prioritized in global palliative care models.

Healthcare staff's intrinsic motivation and professional commitment were repeatedly highlighted as critical facilitators in palliative care utilization [24], which is a similar finding to a study done in Ghana [44], and East Africa [45]. These examples underscore the importance of addressing healthcare provider motivation through both financial and non-financial means to improve healthcare delivery in Ethiopia and similar contexts.

Integrating palliative care into educational curricula is essential for developing a skilled healthcare workforce capable of meeting the complex needs of patients with life-limiting illnesses [26]. In Ethiopia, initial efforts have been made to incorporate palliative care content into some postgraduate and diploma-level programs [24]. However, the health education system struggles to integrate palliative care into undergraduate programs, highlighting the need for curriculum reform to ensure it aligns with real-world clinical demands, despite limited efforts in this area. Similar challenges and developments have been noted in international contexts, including studies from the Netherlands [41], Colombia [42], and India [43]. Leaders focus on diploma and postgraduate courses, requiring curriculum reforms to align palliative care education with real-world clinical practices.

Holistic care models, exemplified by institutions like Hospice Ethiopia, are pivotal in delivering comprehensive palliative care that addresses the multifaceted needs of patients through the integration of medical, psychosocial, and spiritual services, ensuring patient-centered care that resonates with cultural and individual values [14]. The finding is similar to the study done in India [46], Africa [47], and Australia [48]. These international examples underscore the importance of integrating holistic care models into palliative care services. By addressing the medical, psychosocial, and spiritual needs of patients, these models ensure comprehensive care that enhances the quality of life for individuals facing serious illnesses.

Stakeholder involvement is a crucial facilitator in advancing palliative care in Ethiopia, with growing engagement from NGOs, religious institutions, and policymakers [9,25] were contribute to service delivery, raise awareness, and provide support, particularly in rural and underserved areas. Similar trends are seen globally in India [46] and the UK [36]. These international examples underscore the importance of stakeholder involvement in enhancing the reach and sustainability of palliative care services, ensuring comprehensive and holistic care for patients with life-limiting illnesses.

## Strength and limitation

A major strength of this review lies in its adherence to established methodological frameworks, including PRISMA and GRADE-CERQual, which ensured transparency, rigor, and credibility in synthesis and appraisal. Furthermore, the inclusion of diverse Ethiopian contexts enhanced the transferability of findings across regional and health facility settings. Based on these insights, the study recommends strengthening national palliative care policies, expanding training for healthcare providers, and promoting community awareness initiatives to address misconceptions and improve early utilization. Future research should consider longitudinal and implementation studies to evaluate the impact of targeted interventions on service uptake and patient outcomes. However, the limited number of included studies (n = 6) may affect the transferability of the findings, particularly to underrepresented regions or patient populations. Further research is needed to expand the evidence base and include perspectives from more varied sociocultural and geographic contexts.

## Conclusion

This qualitative systematic review synthesized evidence from six studies to explore the multifaceted barriers and facilitators influencing palliative care utilization in Ethiopia. The findings revealed that sociocultural perceptions, limited awareness, inadequate healthcare infrastructure, lack of policy support, and workforce constraints significantly hinder access to palliative care, while strong familial support, community-based interventions, and integration into existing health systems emerged as key facilitators. To address these barriers, we recommend strengthening policy integration of palliative care, expanding training programs for healthcare providers, improving access to essential medications, and promoting community awareness. Culturally sensitive home-based care models and faith-based collaborations should also be explored.

## Supporting information

**S1 Appendix. Detailed search strings.**
(DOCX)

**S1 File. List of identified studies.**
(DOCX)

## Author contributions

**Conceptualization:** Sadik Abdulwehab.

**Data curation:** Sadik Abdulwehab, Frezer Kedir.

**Formal analysis:** Sadik Abdulwehab, Frezer Kedir.

**Investigation:** Sadik Abdulwehab, Frezer Kedir.

**Methodology:** Sadik Abdulwehab, Frezer Kedir.

**Project administration:** Frezer Kedir.

**Software:** Sadik Abdulwehab.

**Supervision:** Sadik Abdulwehab, Frezer Kedir.

**Validation:** Sadik Abdulwehab, Frezer Kedir.

**Visualization:** Frezer Kedir.

**Writing – original draft:** Sadik Abdulwehab, Frezer Kedir.

**Writing – review & editing:** Sadik Abdulwehab, Frezer Kedir.

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
