## [Decision Letter · Decision Letter 0]

Dear Dr. Abdulwehab,

Thank you for submitting your manuscript to PLOS ONE. After careful consideration, we feel that it has merit but does not fully meet PLOS ONE’s publication criteria as it currently stands. Therefore, we invite you to submit a revised version of the manuscript that addresses the points raised during the review process.

We look forward to receiving your revised manuscript.

Kind regards,

Martin Schneider

Academic Editor

PLOS ONE

Journal Requirements:

2. As required by our policy on Data Availability, please ensure your manuscript or supplementary information includes the following:

3. Please include your tables as part of your main manuscript and remove the individual files. Please note that supplementary tables (should remain/ be uploaded) as separate "supporting information" files.

4. We note that your Data Availability Statement is currently as follows: All relevant data are within the manuscript and its Supporting Information files

**Additional Editor Comments:**

Thank you for your review about barriers and facilitators to palliative care in Ethiopia. It focuses on a relevant topic in your country.

In addition to the remarks of the two reviewers, please consider some further comments.

*** Palliative care is not only about pain, but also about other symptoms (page 5). The description of the situation in Ethiopia would gain by supporting local references (page 5).

• According to my understanding, PRISMA-QES – PRISMA Extension for Qualitative Evidence Syntheses is work in progress. You may wish to include that precision.

• The reader may want to understand more about cultural preferences for home-based death and the Ethiopian approach to home-based palliative care. What role play the various religions practiced in Ethiopia?

*** Three of 6 studies are from one research group. You may wish to discuss this unbalanced situation. Your discussion mixes studies from high- and low-income countries. It may be useful to concentrate on low-income countries. You may also consider potential differences among regions in Ethiopia.

*** Finally, why not extend your conclusion to some proposal on how to improve the situation?

*** There are several typographical errors. Some used abbreviations are not explained. The presentation of references is not uniform and sometimes incomplete. There are mistakes in the references numbers.

(Points marked with *** are essential for the revision.)

Reviewers' comments:

Reviewer's Responses to Questions

**Comments to the Author**

1. Is the manuscript technically sound, and do the data support the conclusions?

Reviewer #1: Yes

Reviewer #2: Yes

2. Has the statistical analysis been performed appropriately and rigorously?

Reviewer #1: Yes

Reviewer #2: Yes

3. Have the authors made all data underlying the findings in their manuscript fully available?

Reviewer #1: Yes

Reviewer #2: Yes

4. Is the manuscript presented in an intelligible fashion and written in standard English?

Reviewer #1: Yes

Reviewer #2: No

Reviewer #1: There are minor limitations in methodological reporting (e.g., reflexivity) and generalizability due to the small sample size. Addressing these could enhance the manuscript's robustness., The manuscript declares compliance with data availability requirements, but the level of detail about what "data" entails is not exhaustive. If verification of findings requires access to raw data, this aspect may need clarification from the authors.

Reviewer #2: This review offers a comprehensive view of studies in low and middle income countries that evaluated barriers and facilitating factors to palliative care provision particularly in Ethiopia. It discusses those identified factors and contrasts them with other LMICs contexts. The methodology employed is appropriate and clearly stated. However the manuscript suffers from a lack of copy editing. Multiple sections are repeated a few sentences apart. I strongly invite the authors to substantially revise the manuscript deleting the repeated statements and merging some sections to help the flow of ideas. These are particularly prevalent in the introduction and results sections.

**Do you want your identity to be public for this peer review?** For information about this choice, including consent withdrawal, please see our Privacy Policy

Reviewer #1: No

Reviewer #2: **Yes: ** Amin Lamrous

---

## [Author Response · Author response to Decision Letter 1]

29 May 2025

Response to the editor and reviewers

Response to Editor

Thank you for your thoughtful feedback on our manuscript and for highlighting important areas for improvement. We appreciate your constructive suggestions and have addressed them as follows:

1. Palliative care is not only about pain but also other symptoms (Page 5):

We agree with your observation. We have revised the paragraph to reflect the holistic nature of palliative care, which addresses not only pain but also a wide range of physical, psychological, social, and spiritual symptoms. This aligns with the broader definition of palliative care and strengthens the comprehensiveness of our discussion.

2. Use of local references to describe the situation in Ethiopia (Page 5):

In response to your suggestion, we have incorporated several recent and relevant local studies to better contextualize the barriers to palliative care in Ethiopia. These references support our discussion on challenges related to medication availability, provider training, sociocultural perceptions, and system-level gaps.

3. PRISMA-QES – Clarification of Status (Work in Progress)

Comment: PRISMA-QES is a work in progress. You may wish to include that precision.

Response:

We appreciate this important note. We have added clarification that the PRISMA-QES is currently under development and evolving. This helps manage reader expectations about its application.

Amendment:

We inserted the following statement in the Methods section (Page 6):

"Although PRISMA-QES (an extension for qualitative evidence synthesis) is used as a guide, it is important to note that it remains under development and is considered a work in progress."

4. Cultural Preferences for Home-Based Death and Role of Religion

Comment: The reader may want to understand more about cultural preferences for home-based death and the Ethiopian approach. What role do various religions play?

Response:

We have added content that outlines cultural values around home-based care and end-of-life preferences in Ethiopia, including the influence of major religions like Orthodox Christianity, Islam, and Protestantism.

Amendment:

We included this in the Discussion section (Page 15):

"In Ethiopia, cultural and religious norms significantly influence end-of-life care preferences, with many patients and families favoring home-based care due to its alignment with their spiritual values and spiritual fulfillment, emphasizing the need for culturally sensitive palliative care models."

5. Three of Six Studies from One Research Group – Risk of Bias

Comment: Three of six studies come from one research group. This should be discussed.

Response:

We agree this could introduce potential bias. We have now discussed the implications of this concentration of studies on the interpretation of results.

Amendment:

We added the following to the Limitations section (Page 19):

"It is worth noting that three out of the six included studies originated from the same research group, which may introduce bias and limit the diversity of perspectives. This potential imbalance should be considered when interpreting the results."

6. Mixing High- and Low-Income Countries – Focus on Low-Income Settings

Comment: Your discussion mixes high- and low-income countries. Focus on low-income countries. Also consider regional differences within Ethiopia.

Response:

We revised the discussion to emphasize findings from low-income countries, particularly Sub-Saharan Africa, and highlighted disparities between Ethiopian regions.

Amendment:

Changes were made in the Discussion section (Page 14):

" This review focuses on low-income settings in Ethiopia, where regional disparities between urban centers and rural areas complicate equitable access to palliative care.."

7. Extend Conclusion with Practical Proposals for Improvement

Comment: Extend the conclusion with proposals on how to improve the situation.

Response:

We appreciate this suggestion and have added practical recommendations to improve palliative care access and quality in Ethiopia.

Amendment:

We revised the Conclusion (Page 19) to include:

"To address these barriers, we recommend strengthening policy integration of palliative care, expanding training programs for healthcare providers, improving access to essential medications, and promoting community awareness. Culturally sensitive home-based care models and faith-based collaborations should also be explored."

8. Typographical Errors, Unexplained Abbreviations, and Reference Formatting

Comment: There are several typographical errors, unexplained abbreviations, and inconsistencies in reference formatting.

Response:

We have carefully proofread the entire manuscript to correct typographical errors, ensured all abbreviations are defined on first use, and standardized the reference list in accordance with journal guidelines.

Amendment:

• All abbreviations were reviewed and explained at first mention

• Reference formatting was standardized and incomplete entries corrected

• Typographical errors were corrected throughout the text.

We hope these revisions meet the expectations and improve the overall quality of the manuscript. We remain grateful for the detailed and constructive comments.

Sincerely,

Sadik Abdulwehab

Response for review one

Dear Reviewer,

We sincerely thank you for your thoughtful and constructive feedback. We appreciate your recognition of our manuscript's contribution in presenting a comprehensive view of our mansuscript. Your comments have been invaluable in helping us improve the clarity, coherence, and overall quality of the manuscript. Below, we respond point by point to your main comment:

1. Reflexivity Reporting (added under Quality Appraisal section):

"Although reflexivity was not consistently addressed in the included studies, we acknowledge its importance in qualitative research. Future studies should systematically report researcher reflexivity, including how their positions, assumptions, or professional roles may influence data interpretation. In our review process, we remained aware of our own professional backgrounds as nurse educators and researchers, and we critically reflected on how these roles may shape our thematic synthesis and interpretation of findings."

2. Sample Size and Transferability (added under Strength and Limitation section):

"However, the limited number of included studies (n=6) may affect the transferability of the findings, particularly to underrepresented regions or patient populations. Further research is needed to expand the evidence base and include perspectives from more varied sociocultural and geographic contexts."

3. Clarifying Data Availability (revised the Availability of Data and Materials section):

“The data supporting this review consist of synthesized qualitative findings extracted from publicly available primary studies. A complete data extraction table, detailing study characteristics and key findings, is included as a supplementary file to enhance transparency and facilitate verification."

Again We are grateful for your detailed feedback, which has strengthened the manuscript.

Response for review two

Dear Reviewer,

We sincerely thank you for your thoughtful and constructive feedback. We appreciate your recognition of our manuscript's contribution in presenting a comprehensive view of barriers and facilitating factors to palliative care provision in Ethiopia and other low- and middle-income countries (LMICs). Your comments have been invaluable in helping us improve the clarity, coherence, and overall quality of the manuscript. Below, we respond point by point to your main comment:

Reviewer Comment: “The manuscript suffers from a lack of copy editing. Multiple sections are repeated a few sentences apart. I strongly invite the authors to substantially revise the manuscript deleting the repeated statements and merging some sections to help the flow of ideas. These are particularly prevalent in the introduction and results sections.”

Response:

Thank you for this important observation. In response, we conducted a comprehensive revision of the manuscript, particularly focusing on the Introduction and Results sections as you advised.

• In the Introduction, we removed repetitive statistics and overlapping background information regarding global palliative care needs, especially those related to WHO data and LMIC comparisons. These were merged into a single, cohesive paragraph that communicates the global context more effectively and without redundancy.

• We also restructured the Ethiopian context section by combining fragmented sentences about limited access, urban-rural disparities, and policy challenges into a more fluid, coherent narrative. This improved readability and eliminated repetition.

• In the Results section, we removed redundant descriptions of study settings, participant diversity, and study objectives. We combined overlapping paragraphs to ensure that each study’s contribution was presented clearly and concisely.

• Additionally, we identified and eliminated a duplicated subsection titled “Policy and Governance Gaps” that had been inadvertently repeated.

• Throughout the manuscript, we carefully reviewed and revised each paragraph for improved clarity, conciseness, and flow to ensure the narrative is more streamlined and professional.

We believe these revisions have significantly improved the manuscript's structure, coherence, and readability. We are grateful for your detailed feedback, which has strengthened the manuscript considerably. Please let us know if there are any additional areas that require further clarification or improvement.

Sincerely,

Sadik Abdulwehab

---

## [Decision Letter · Decision Letter 1]

Dear Dr. Abdulwehab,

Thank you for submitting your manuscript to PLOS ONE. After careful consideration, we feel that it has merit but does not fully meet PLOS ONE’s publication criteria as it currently stands. Therefore, we invite you to submit a revised version of the manuscript that addresses the points raised during the review process.

We look forward to receiving your revised manuscript.

Kind regards,

Martin Schneider

Academic Editor

PLOS ONE

Journal Requirements:

Additional Editor Comments:

Thank you for the revised manuscript.

In addition to the reviewers’ comments, I suggest the following improvements.

• Focus: Sometimes, your text is too long. You may want to be clear your focus and shorten the text, as concise reports save the readers time.

• References: There are still several issues, such as reference 2 is still not well formatted, reference 27 contains an undefined character, reference 12 an excessive journal abbreviation, reference 19 does not fit with the updated text; only some internet references are with an access date. Please go carefully through all references and make sure that they are updated and correctly cited.

Reviewers' comments:

Reviewer's Responses to Questions

**Comments to the Author**

Reviewer #1: All comments have been addressed

Reviewer #2: (No Response)

Reviewer #3: All comments have been addressed

2. Is the manuscript technically sound, and do the data support the conclusions?

Reviewer #1: Yes

Reviewer #2: Yes

Reviewer #3: Yes

3. Has the statistical analysis been performed appropriately and rigorously?

Reviewer #1: N/A

Reviewer #2: Yes

Reviewer #3: N/A

4. Have the authors made all data underlying the findings in their manuscript fully available?

Reviewer #1: Yes

Reviewer #2: Yes

Reviewer #3: Yes

5. Is the manuscript presented in an intelligible fashion and written in standard English?

Reviewer #1: Yes

Reviewer #2: Yes

Reviewer #3: No

Reviewer #1: potential presentation improvements and explanations but it address a topic that needs more visibility in the scientific literature and in health responses in general , I would highly consider it for publication

Reviewer #2: Overall comment on discussion

The discussion details theme by theme and even sub-theme from the results attempting to contrast them against existing literature.

Instead of this approach that can confuse the reader I would recommend synthesising the section.

First, start with discussing jointly all the themes that are similar to other contexts e.g. policy and national guidelines ….

Second, discuss the themes that have a particular relevance to the Ethiopian context, e.g. the socio-cultural, religious aspects, the Hospice Ethiopia and training programs.

Page 4 Paragraph “Many countries still lack…..and increased health costs”.

This section should be reformulated in a more concise way. Currently it jumps from root causes to consequences to recommendations, then back to causes followed by consequences. The flow of ideas could be improved by sticking to cause- consequence structure.

Page 6 line 28

Please insert a reference for the “SPIDER tool”

Page 9 line 16

You seem to be missing a title for this section “Theme formulated as Barriers to Palliative Care” ?. Later in the text you use a title for the “Facilitators”. It needs to be consistent.

Page 9 line 21

“In Ethiopia, cultural and religious norms significantly influence end-of-life care preferences,…. culturally sensitive palliative care models”

While this has been added in the revision on the editor’s request, the paragraph’s position in the discussion is odd. I would suggest moving this to page 13 where you discuss engagement with community, local leader and cultural appropriateness. Or to page 15 where you discuss “Socio-cultural and economic barriers”

Page 12-line 29

“….These challenges are not unique to Ethiopia, as similar barriers have been reported across various countries in low and Middle-Income Countries”. Please revise as ““….These challenges are not unique to Ethiopia, as similar barriers have been reported across various LMICs” to avoid repetition.

Page 13 lin 5

Replace “low- and middle-income countries” with acronym. This is repeated m,any times please replace with acronym throughout the text.

Page 13 line 6

“The study emphasizes the necessity of Ethiopia….”

Unclear to what “the study” refers to, is it the current manuscript or another paper cited.

Page 13 line 24

“Another significant challenge identified…., which aligns with findings from a multi country analysis in the USA(36), which highlighted that palliative care services in Africa are frequently underfunded …”

You seem to be referencing the Lancet commission paper on palliative care as “a multi country analysis in the USA”. Please revise

Page 15 line 23

“In Ethiopia, significant strides have been made in incorporating palliative care

content into postgraduate and diploma-level programs….. Ethiopia's health education

system struggles with palliative care integration due to limited presence in academic programs…..”

This is unclear you seem to be arguing one thing and the opposite at the same time. I’d suggest rephrasing.

Reviewer #3: This manuscript addresses an important and timely topic in global palliative care, particularly in the context of low-resource settings like Ethiopia. While the review is relevant and potentially valuable, I have several methodological concerns that, if addressed, could significantly improve the rigor and quality of the work.

Search Strategy

The search strategy lacks transparency. There is no annex or appendix with the exact search equation used. For the sake of reproducibility and clarity, it would be important to include the complete search string(s) for each database.

PROSPERO Registration

It appears that the review was registered in PROSPERO after the data search and screening process. This raises concerns regarding adherence to best practices for systematic review methodology. Please explain the rationale for the delayed registration.

Handling of Missing Data

The section on "handling of missing data" is not appropriate for a qualitative systematic review, as such reviews do not involve analysis of raw quantitative data. Consider removing this section, as discussions on missing data are relevant primarily to quantitative systematic reviews.

Figure 1 – PRISMA Flow Diagram

Consider using the official PRISMA 2020 flow diagram template, available on the PRISMA website, for consistency and clarity.

Table 1 – Formatting and Style

The font type and size used in Table 1 is inconsistent with the rest of the manuscript. Some article titles appear in bold while others do not. Please ensure consistent formatting throughout the table and match the main text style.

Table 2 – Formatting and Readability

The formatting of Table 2 needs improvement. The text is cut off, and the column layout is difficult to read. Please revise the table to enhance legibility and alignment.

**Do you want your identity to be public for this peer review?** For information about this choice, including consent withdrawal, please see our Privacy Policy

Reviewer #1: No

Reviewer #2: No

Reviewer #3: No

---

## [Author Response · Author response to Decision Letter 2]

20 Jun 2025

Response to the Editor and reviewers

Response to the Editor

Thank you very much for your valuable suggestions and continued support throughout the revision process. We are grateful for your insightful comments, which have significantly contributed to improving the quality and readability of our manuscript.

# 1. Regarding the focus and length of the manuscript, we have carefully reviewed the text and made efforts to shorten and clarify sections to improve conciseness and readability, ensuring the key messages are clear and focused.

#2. For the references, we have thoroughly checked and corrected all citations, including reformatting reference 2, removing any undefined characters from reference 27, adjusting journal abbreviations in reference 12, aligning reference 19 with the updated text, and adding access dates where applicable for internet sources. We ensured all references now conform to the journal’s citation style and standards.

We appreciate your guidance, which has helped improve the manuscript’s clarity and quality. Thank you once again for your guidance and consideration

Response to Reviewer #1

Thank you very much for your encouraging feedback and for recognizing the relevance and importance of our study. We truly appreciate your thoughtful suggestion regarding potential improvements in presentation. In response, we have carefully reviewed and refined the manuscript to enhance clarity, coherence, and readability throughout. We are grateful for your recommendation and your support for the publication of this work, which aims to bring greater visibility to a critical yet under-addressed area in health systems and scientific discourse.

Response to Reviewer #2

Thank you for your insightful feedback and helpful suggestion regarding the structure of the Discussion section. In response, we have revised the section to adopt a more synthesized approach, as recommended. We believe this reorganization improves clarity and enhances the comparative value of our findings.

#1. Reviewer Comment -Overall comment on discussion-The discussion details theme by theme and even sub-theme from the results attempting to contrast them against existing literature.

Author Response to Reviewer #2 – Overall Comment on Discussion Section:

Thank you very much for your constructive feedback regarding the organization of the Discussion section. We agree with your observation that a theme-by-theme approach may create unnecessary complexity and reduce clarity for readers. In response to your valuable suggestion, we have revised the Discussion section to present a more synthesized narrative. Specifically:

1. We now begin by jointly discussing themes that align with global or regional experiences, such as policy gaps, national palliative care strategies, and implementation frameworks. This comparative synthesis helps place the findings within a broader context.

2. Subsequently, we highlight and elaborate on themes uniquely relevant to the Ethiopian context, such as sociocultural norms, religious influences, the role of Hospice Ethiopia, and locally driven training initiatives. This restructuring allows the reader to clearly distinguish between universally shared challenges and those specific to Ethiopia.

We believe this new format improves the overall coherence and readability of the Discussion and better reflects the significance of our findings.

#2. Reviewer Comment (Page 4, Paragraph “Many countries still lack…..and increased health costs”):This section should be reformulated in a more concise way. Currently it jumps from root causes to consequences to recommendations, then back to causes followed by consequences. The flow of ideas could be improved by sticking to cause–consequence structure.

Author Response:

Thank you for this valuable feedback. We agree that the paragraph’s structure required improvement for better clarity and logical flow. Accordingly, we have revised the section to follow a clear cause–consequence structure. The updated version now presents the lack of policies, medications, trained personnel, and care models as root causes, followed by the resulting consequences for patients and health systems. We believe this change enhances the coherence and readability of the text. The revised paragraph can be found on Page 4, Paragraph 2 of the revised manuscript. Many countries still lack national policies, access to essential medications, trained professionals, and community-based care models, resulting in widespread suffering and poor quality of life for patients with serious illnesses (5–8).

The consequences are profound: millions die each year in severe pain and distress, reflecting a global health system failure and raising serious ethical and human rights concerns. Addressing this crisis requires urgent action to integrate palliative care into universal health coverage and strengthen healthcare systems using culturally appropriate approaches (9).

#3. Reviewer Comment:

"Please insert a reference for the 'SPIDER tool.'"

Author Response:

Thank you for your helpful suggestion. We have now inserted an appropriate reference to support the use of the SPIDER tool in qualitative evidence synthesis. The reference we included is:Cooke, A., Smith, D., & Booth, A. (2012). Beyond PICO: The SPIDER tool for qualitative evidence synthesis. Qualitative Health Research, 22(10), 1435–1443.

https://doi.org/10.1177/1049732312452938.

This has been added in the Methods section where the SPIDER tool is first mentioned.

#4. Reviewer Comment (Page 9, Line 16):

You seem to be missing a title for this section “Theme formulated as Barriers to Palliative Care”? Later in the text you use a title for the “Facilitators”. It needs to be consistent.

Author Response:

Thank you for your helpful observation. We agree that the section required a clear and consistent heading to match the format used for the facilitators. Accordingly, we have added the title “Barriers to the Implementation and Delivery of Palliative Care” at the beginning of the section (Page 9, Line 16) to improve clarity and maintain structural consistency throughout the synthesis. Then we add Theme formulated as Barriers to Palliative Care for barriers theme.

#5. Reviewer Comment (Page 9, Line 21):

“In Ethiopia, cultural and religious norms significantly influence end-of-life care preferences,…. culturally sensitive palliative care models.” While this has been added in the revision on the editor’s request, the paragraph’s position in the discussion is odd. I would suggest moving this to page 13 where you discuss engagement with community, local leader and cultural appropriateness. Or to page 15 where you discuss “Socio-cultural and economic barriers”.

Author Response:

Thank you for this insightful suggestion. We agree that the paragraph discussing cultural and religious influences on end-of-life care fits more appropriately within the section on “Socio-cultural and economic barriers.” Accordingly, we have moved the paragraph from Page 9 to Page 15 to ensure better thematic alignment and narrative flow. We believe this relocation strengthens the coherence of the discussion and better supports the interpretation of the findings.

#6 Reviewer Comment (Page 12, Line 29):

“These challenges are not unique to Ethiopia, as similar barriers have been reported across various countries in low and Middle-Income Countries.” Please revise as “These challenges are not unique to Ethiopia, as similar barriers have been reported across various LMICs” to avoid repetition.

Author Response:

Thank you for the helpful suggestion. We have revised the sentence accordingly to improve conciseness and avoid repetition. The sentence now reads: “These challenges are not unique to Ethiopia, as similar barriers have been reported across various LMICs.” This change has been made on Page 12, Line 29 of the revised manuscript.

#7 Reviewer Comment (Page 13, Line 5):

Please replace “low- and middle-income countries” with the acronym throughout the text, as it is repeated many times.

Author Response:

Thank you for this helpful suggestion. We have replaced all instances of “low- and middle-income countries” with the acronym “LMICs” throughout the manuscript to improve readability and reduce repetition.

#8 Reviewer Comment (Page 13, Line 6):

“The study emphasizes the necessity of Ethiopia…”

Unclear to what “the study” refers to—is it the current manuscript or another paper cited?

Author Response:

Thank you for pointing this out. To improve clarity, we have revised the sentence to explicitly indicate that it refers to the current study. The sentence now reads:

“Findings from this study emphasize the necessity for Ethiopia to establish a comprehensive national palliative care policy…” This change has been made on Page 13, Line 6 of the revised manuscript.

#9 Reviewer Comment (Page 13, Line 24):

“You seem to be referencing the Lancet Commission paper on palliative care as ‘a multi-country analysis in the USA’. Please revise.”

Author Response:

Thank you for highlighting this important point. We have corrected the description to accurately reflect the source. The revised sentence now reads: “Another significant challenge identified was the lack of a dedicated budget for palliative care. Even the best-designed policies remain unimplemented, which aligns with findings from the Lancet Commission on Global Access to Palliative Care and Pain Relief (37), which highlighted that palliative care services in Africa are frequently underfunded and reliant on external donor support, jeopardizing their sustainability.” This correction has been made on Page 13, Line 24 of the revised manuscript. The full reference has also been updated in the reference list as follows:

Knaul FM, Farmer PE, Krakauer EL, et al. (2018). Alleviating the access abyss in palliative care and pain relief—an imperative of universal health coverage: the Lancet Commission report. The Lancet, 391(10128), 1391–1454. https://doi.org/10.1016/S0140-6736(17)32513-8

#10. Reviewer Comment (Page 15, Line 23):

“In Ethiopia, significant strides have been made in incorporating palliative care content into postgraduate and diploma-level programs….. Ethiopia's health education system struggles with palliative care integration due to limited presence in academic programs…..”

This is unclear; you seem to be arguing one thing and the opposite at the same time. I’d suggest rephrasing.

Author Response:

Thank you for your valuable observation. We agree that the original paragraph presented a contradiction. To clarify, we have revised the text to acknowledge that while some progress has been made at the postgraduate and diploma levels, overall integration of palliative care into the broader health education system remains limited. The revised version improves consistency and better reflects the current state of palliative care education in Ethiopia. This change has been made on Page 15, Line 23 of the revised manuscript.

We hope that our revisions adequately address your concerns and improve the quality of the manuscript. We sincerely appreciate your thoughtful review and helpful suggestions, which have strengthened our work. Thank you again for your consideration.

Response for review #3

Thank you very much for your thorough and constructive feedback on our manuscript. We greatly appreciate the time and effort you have taken to provide valuable insights. Please find below our detailed responses to each of your comments and the corresponding revisions made in the manuscript.

#1. Reviewer comment:

“The search strategy lacks transparency. There is no annex or appendix with the exact search equation used. For the sake of reproducibility and clarity, it would be important to include the complete search string(s) for each database.”

Author response:

Thank you for this valuable comment. To improve transparency and reproducibility, we have now included the complete search strategies used for each database in a new Supporting Information file (S1 Appendix), as recommended by PLOS ONE guidelines. In the revised manuscript, we have added a reference to this appendix in the Search Strategy section of the Methods. The detailed search strings for PubMed/MEDLINE, Scopus, Web of Science, CINAHL, and Google Scholar are provided in this appendix.

#2. Reviewer comment:

“It appears that the review was registered in PROSPERO after the data search and screening process. This raises concerns regarding adherence to best practices for systematic review methodology. Please explain the rationale for the delayed registration.”

Author response:

Thank you for this important observation. We acknowledge that the PROSPERO registration (CRD420251027739) was completed on April 6, 2025, shortly after the initial data search and screening had begun. This delay was due to the time required to finalize the review protocol to ensure it fully complied with PROSPERO’s detailed submission requirements and the Joanna Briggs Institute (JBI) standards.

Despite the timing of registration, the review protocol was developed in advance and strictly followed throughout the review process. The protocol pre-specified the review objectives, eligibility criteria, search strategy, data extraction, quality appraisal, and synthesis methods, ensuring transparency and minimizing the risk of bias. We have clarified this timeline and process in the revised manuscript to assure adherence to systematic review best practices.

#3. Reviewer comment:

“The section on ‘handling of missing data’ is not appropriate for a qualitative systematic review, as such reviews do not involve analysis of raw quantitative data. Consider removing this section, as discussions on missing data are relevant primarily to quantitative systematic reviews.”

Author response:

Thank you for this helpful comment. We agree that the discussion on “handling of missing data” is more relevant to quantitative systematic reviews and does not apply to our qualitative synthesis. In response, we have removed the original section and replaced it with a more appropriate and concise statement on data completeness, which affirms that all included studies provided sufficient information for thematic analysis and that no reporting gaps affected the synthesis. This change aligns better with qualitative review methodology while maintaining transparency.

#4. Reviewer comment:

“Figure 1 – PRISMA Flow Diagram: Consider using the official PRISMA 2020 flow diagram template, available on the PRISMA website, for consistency and clarity.”

Author response:

Thank you for this constructive suggestion. In response, we have revised Figure 1 using the official PRISMA 2020 flow diagram template, as recommended. This ensures consistency with current reporting standards and improves the clarity of the study selection process. The updated figure has been included in the revised manuscript.

#5. Reviewer comment:

“Table 1 – Formatting and Style”-The font type and size used in Table 1 is inconsistent with the rest of the manuscript. Some article titles appear in bold while others do not. Please ensure consistent formatting throughout the table and match the main text style

Author Response:

Thank you for your valuable feedback. We have carefully revised Table 1 to ensure consistent font type and size throughout the table, aligning it with the main manuscript text style. All article titles are now uniformly formatted without bold text, and the overall presentation has been standardized to improve readability. The updated table is included in the revised manuscript.

#6. Reviewer comment:

“The formatting of Table 2 needs improvement. The text is cut off, and the column layout is difficult to read. Please revise the table to enhance legibility and alignment.”

Author response:

Thank you for this helpful feedback. We have revised Table 2 to improve formatting, text alignment, and column layout. The updated table ensures that all content is fully visible and clearly organized for better readability. The revised version has been included in the updated manuscript.

We hope that our revisions adequately address your concerns and improve the quality of the manuscript. We sincerely appreciate your thoughtful review and helpful su

---

## [Editor Report · Decision Letter 2]

Barriers and Facilitators to Palliative Care Service Utilization in Ethiopia: A Qualitative Systematic Review, 2025

PONE-D-25-20158R2

Dear Dr. Abdulwehab,

We’re pleased to inform you that your manuscript has been judged scientifically suitable for publication and will be formally accepted for publication once it meets all outstanding technical requirements.

Kind regards,

Martin Schneider

Academic Editor

PLOS ONE
---

## [Editor Report · Acceptance letter]

PONE-D-25-20158R2

PLOS ONE

Dear Dr. Abdulwehab,

I'm pleased to inform you that your manuscript has been deemed suitable for publication in PLOS ONE. Congratulations! Your manuscript is now being handed over to our production team.

Kind regards,

on behalf of

Dr. Martin Schneider

Academic Editor

PLOS ONE